# Advances in the Diagnosis and Therapeutic Management of Gastroenteropancreatic Neuroendocrine Neoplasms (GEP-NENs)

**DOI:** 10.3390/cancers14082028

**Published:** 2022-04-17

**Authors:** Krzysztof Kaliszewski, Maksymilian Ludwig, Maria Greniuk, Agnieszka Mikuła, Karol Zagórski, Jerzy Rudnicki

**Affiliations:** Department of General, Minimally Invasive and Endocrine Surgery, Wroclaw Medical University, Borowska Street 213, 50-556 Wroclaw, Poland; max.ludwig209@gmail.com (M.L.); greniukmarysia@gmail.com (M.G.); agnieszka.mikula.99@gmail.com (A.M.); zagorskikarol@op.pl (K.Z.); jerzy.rudnicki@umw.edu.pl (J.R.)

**Keywords:** GEP-NENs, GEP-NETs, laboratory diagnostic, NETest, imaging, nuclear medicine, SPECT, SSA, radionuclide treatment, immunotherapy, chemotherapy, surgery

## Abstract

**Simple Summary:**

Gastroenteropancreatic neuroendocrine neoplasms (GEP-NENs) are a group of tumors with different clinical manifestations, various localizations in the human body, and a particularly wide range of histological types, grades and severities. Their diagnosis and therapeutic management are complex. Current standards are not always effective, and sometimes require modifications. Thus, knowledge about GEP-NENs and approaches for the treatment of these patients in Europe and all over the world are constantly evolving. The aim of this review is to highlight the progress in diagnostics and treatment of GEP-NENs. Special attention is given to GEP-NETs as the most widely reported in the literature.

**Abstract:**

Neuroendocrine neoplasms (NENs) are an increasingly common cause of neoplastic diseases. One of the largest groups of NENs are neoplasms localized to the gastroenteropancreatic system, which are known as gastroenteropancreatic NENs (GEP-NENs). Because of nonspecific clinical symptoms, GEP-NEN patient diagnosis and, consequently, their treatment, might be difficult and delayed. This situation has forced researchers all over the world to continue progress in the diagnosis and treatment of patients with GEP-NENs. Our review is designed to present the latest reports on the laboratory diagnostic techniques, imaging tests and surgical and nonsurgical treatment strategies used for patients with these rare neoplasms. We paid particular attention to the nuclear approach, the use of which has been applied to GEP-NEN patient diagnosis, and to nonsurgical and radionuclide treatment strategies. Recent publications were reviewed in search of reports on new strategies for effective disease management. Attention was also paid to those studies still in progress, but with successful results. A total of 248 papers were analyzed, from which 141 papers most relevant to the aim of the study were selected. Using these papers, we highlight the progress in the development of diagnostic and treatment strategies for patients with GEP-NENs.

## 1. Introduction

Gastroenteropancreatic neuroendocrine neoplasms (GEP-NENs) are a diverse group of rare gastrointestinal tumors derived from neuroendocrine cells, which form a diffuse neuroendocrine system (DES) [1,2]. Cells of the DES are scattered throughout the body and secrete amine- and hormone-like molecules, mainly peptide hormones, as a result of nerve stimulation [1]. GEP-NETs account for approximately 2/3 of the total number of all neuroendocrine tumors (NETs) and approximately 2% of all GEP-NENs [2]. They can originate from the stomach, intestines (from duodenum to rectum) and pancreas.

### 1.1. Epidemiology

The overall incidence of this type of tumor is low. The occurrence is less than 10/100,000 per year and exhibits a fair degree of variation across countries [3]. Studies show that there has been a large increase in the incidence of this group of diseases over the last 30 and even 40 years [4,5,6]. For the most part, this increased incidence is associated with the detection of smaller or asymptomatic lesions [4,7]. The findings of some studies indicated a 6.4-fold increase in incidence from 1973 to 2012 [7]. The Massironi et al. [8] results indicated an increased incidence of lesions detected in autopsy examinations, which was 2–5 times higher than the estimates reported for the general patient population, for which the average incidence was 2.5–5/100,000 per year.

Most GEP-NENs localize to the small intestine (31%) and rectum (29%) [9]. In the U.S. from 1975 to 2015, age-adjusted incidence increased significantly, particularly for low-grade lesions (GEP-NETs Grade 1), localized lesions, and rectal lesions. Age, sex, marital status, tumor size, grade, stage and site were significantly associated with the overall survival of patients with GEP-NETs [6]. Notably, the proportion of localized GEP-NETs and rectal lesions increased. The mean age of patient diagnosis increased by 9 years [6]. There is a slightly higher incidence in men than in women (5.35 vs. 4.76/100 000 per year) [2], but some studies do not confirm it [10].

### 1.2. Classification

The International Agency for Research on Cancer—World Health Organization (IARC-WHO) in 2017 proposed the division of NENs into well-differentiated NETs and poorly differentiated neuroendocrine cancers (NECs). The division is based on molecular differences: the typical genes in which mutations occur for NETs are MEN1, DAXX, and ATRX (they are entity-defining), and for NECs, they are TP53 and RB1. This distinction was repeated in the 2019 WHO Classification Guidelines for tumors of the digestive system [11].

The exact etiology and classification of GEP-NETs depends on the site of origin of the disease [12]. The subtypes of GEP-NENs based on the 2019 WHO classification scheme are included in Table 1.

Approximately 20% of NETs occur as a part of a genetic syndrome [13]. In Table 2, we show selected genetic diseases associated with an increased prevalence of GEP-NETs. The various etiologies are associated with different clinical courses, ranging from slow progression to a very aggressive disease course, but chronic courses predominate, especially in patients with small lesions [14]. Some of these tumors secrete increased amounts of hormones and amines that cause symptoms (e.g., insulin-secreting insulinoma causes transient episodes of hypoglycemia) and cause additional clinical and diagnostic problems. For this reason, the classification scheme of NETs distinguishes between functioning subtypes, in which secretions cause symptoms, and nonfunctioning subtypes, in which secretions do not cause any symptoms [15].

### 1.3. Prognosis

The prognosis is highly variable. For patients with localized tumors, the mean 5-year survival is 97%, which is much higher than those for regionally and extensively spread tumors, which are 81% and 39%, respectively. The 5-year survival for patients with tumors originating from the pancreas is 52%; from the stomach, 82%; the small intestine, 84%; the colon, 62%; the appendix, 88%; and the rectum, 96%. Worse survival is also associated with higher age; those aged younger than 30 years have a 5-year survival of 92%, those aged between 30 and 60 years have a 5-year survival of 87%, and those aged older than 60 years have a 5-year survival 72%. A higher tumor grade is also highly associated with a lower 5-year survival: that of patients with grade 1 tumors is 91%, the values are 78% for patients with grade 2 tumors, 21% for patients with grade 3 tumors, and 21% for patients with grade 4 tumors [6].

## 2. Laboratory Diagnostics

Both nonspecific and specific markers are applicable in the laboratory diagnosis of patients with GEP-NENs. For hormonally active tumors, it is possible to measure the concentrations of specific products secreted by their cells. These include, among others, various peptides and biogenic amines depending on the tumor type. These molecules are helpful in patient diagnosis and in monitoring therapy; however, their use has significant limitations [2]. As monoanalytical markers, they reflect the secretory activity of the tumor to the exclusion of assessing its biology. Moreover, only approximately 50% of NENs exhibit sufficiently high secretory activity to allow for the detection of specific markers [16,17]. Among nonspecific markers, chromogranin A (CgA), neurospecific enolase, pancreatic polypeptide, and synaptophysin are examples [2,18]. The greatest importance in the laboratory diagnosis of patients is attributed to the determination of CgA concentration in serum or plasma. It is helpful in cancer diagnosis, in monitoring treatment and the natural course of the disease, and as a prognostic marker of survival [2]. However, there are some limitations in the utility of this type of marker. Its sensitivity is dependent on the grade and extent of the tumor. As a volumetric marker, it reflects tumor size, so it may result in false negativity in patients with small lesions [19]. CgA is also secreted by healthy tissues, and its concentration may also increase in the case of active inflammation in the body. The use of proton pump inhibitors is another factor causing false positives [20]. CgA levels are rarely elevated in patients with NETs of the colon and rectum, so CgA measurement is not diagnostically applicable in these patients [21]. Due to the multitude of limitations of the best marker currently available, the laboratory diagnosis of patients with NETs needs to be improved.

### 2.1. NETest

High hopes have been placed on the use of liquid biopsies in the diagnosis of patients with GEP-NETs. In recent years, there have been many publications of NETest. It is a multianalytical assay that relies on the detection of NET transcripts circulating in the blood [16]. The analysis is based on mRNA detection performed through PCR, and the result is expressed as the NETest score (0–100%) [20,22].

### 2.2. NETest vs. CgA

Van Treijen et al. [20] conducted a study to confirm the discriminatory ability of NETest by comparing it with that of the marker that is currently most widely used, CgA. The results showed a higher sensitivity of the NETest than that of CgA (93% and 56%, respectively) [20]. CgA levels, contrary to previous opinions, mainly reflect the secretory activity of the tumor [22], while in contrast, the results of the NETest are independent of this activity. This difference is probably the reason for the difference in sensitivity. On the other hand, the specificity of NETest in the described work was much lower than that of CgA. Several possible reasons for false positives have been mentioned. Some of the genes whose transcripts are detected by NETest are also overexpressed in conditions other than cancer, for example, during stress or in the presence of inflammation. In addition, the test may result in false positivity if other tumors not classified as NETs, but showing neuroendocrine differentiation, are present. Such a phenomenon may occur in prostate cancer, among others [20]. A few years later, a comparison of the utility of NETest with that of CgA was made using a slightly different approach. In their work, the authors validated NETest as a prognostic marker and showed that it was the strongest predictor of disease progression [19]. Investigators in this cohort study also examined the use of the described diagnostic method as a biomarker to evaluate the effectiveness of surgical treatment for patients with NETs. There is potential for reducing the use of imaging methods in postoperative monitoring. This practice would be associated with a reduction in costs and in patient exposure to harmful radiation. On the other hand, CgA has been shown to be a better marker for predicting mortality [19].

### 2.3. NETest vs. Imaging

In their study, Malczewska et al. [22] analyzed data from a cohort of patients and a healthy control group. The aim was to validate NETest as a tool to diagnose patients with small intestinal NET (SINET) and pancreatic NET (PanNET), and to assess disease status (stable or progressive disease). Imaging studies were performed concurrently with blood sampling for NETest to assess disease status, and lesions were evaluated according to RECIST 1.1 criteria. The results presented in the publication support the effectiveness of NETest as a diagnostic marker for both PanNETs and SINETs. The NETest score was elevated in all patients screened. The NETest results showed high concordance with those of imaging studies, which currently play a key role in the diagnosis of patients with neuroendocrine tumors. In addition, NETest has proven to be more effective than imaging in several cases by showing a positive result even with a concurrent false negative computed tomography (CT) result. NETest has also performed well in assessing disease status. Biopsy samples demonstrated much higher rates of positivity in progressive disease patients than in stable disease ones. Thus, NETest results in this aspect also showed concordance with results based on imaging and status assessment using RECIST 1.1 [22].

### 2.4. NETest vs. PRRT Predictor Quotient (PPQ)

Peptide receptor radionuclide therapy (PRRT) predictor quotient (PPQ) is an index that determines the predicted sensitivity of a tumor to PRRT treatment. Depending on whether the tumor will respond to treatment, i.e., whether the patient is in the responder or non-responder group, the PPQ score will be PPQ-positive or PPQ-negative, respectively [23]. In simple terms, the course of analysis involves performing eight NETests to look for transcripts of specific genes related to growth factor production and metabolism [24]. Based on their expression levels, PPQ is determined. In their work, Bodei et al. [23] verified the utility of NETest in this assessment. For this purpose, they compared the NETest scores obtained before and after the implementation of therapy to the results of imaging studies. A significant decrease in NETest score after treatment of patients in the responder group confirmed its clinical usefulness. Furthermore, it has been shown that NETest results are also affected by minimal tumor growth undetectable by imaging. This finding means that with slow-growing lesions, NETest may be a more sensitive method of assessing treatment response than radiological methods [23]. Öberg et al. [25] reviewed the literature on NETest that was published between 2015 and 2019. Ten publications meeting the criteria were identified. Based on the papers reviewed, the utility of NETest was assessed in four domains: the diagnosis of the disease, determination of disease status (stable or progressive disease), determination of natural history, and assessment of response to treatment. The meta-analysis results showed that NETest meets the criteria of a good biomarker for the mentioned applications. However, it is not suitable as a screening tool. For such purposes, low-complexity tests with high sensitivity are used to identify disease in asymptomatic people. The NETest, as a high-complexity test, is used for people with suspected disease to exclude or confirm it [25].

### 2.5. HOX Gene Expression Analysis

The work of Di Mauro et al. [26] demonstrated the association of *13 HOX* and *HOTAIR* gene expression with the course of neuroendocrine tumorigenesis. Thus, the possibility of their use as prognostic markers of NETs was recognized. Physiologically, *HOX* genes play a role during embryogenesis, whereas the role of lncRNA *HOTAIR* is to modulate their expression. Changes in the expression of *13 HOX A*, *C*, and *D* genes and *HOTAIR* have been shown to correlate with tumor aggressiveness. Downregulation of *HOX* is associated with poorer prognosis. In the case of *HOTAIR*, there is an inverse relationship with patient prognosis. The lower the expression is, the lower the tumor grade, and thus, the better the prognosis of the patient. Determination of *HOX* gene expression levels was performed by immunohistochemical analysis, which revealed the presence of their products. In turn, *HOTAIR* expression was examined by in situ hybridization (ISH) of long noncoding RNAs (lncRNAs). Immunohistochemistry for HOX proteins and the use of *HOTAIR* lncRNA as a circulating biomarker present themselves as promising prognostic methods for the diagnosis of patients with GEP-NETs [26].

### 2.6. Epigenome Analysis

In addition to the genome, various types of epigenetic modifications are also responsible for tumor biology. Rinke et al. [24] suggested the possibility of the DNA methylation analysis of circulating tumor cells (CTCs) for the diagnosis of patients with GEP-NETs. Modifications in specific fragments of genetic material are responsible for differences in disease progression. For example, gastric inhibitory polypeptide receptor (GIPR) methylation is correlated with metastasis. Through epigenome analysis, several groups of GEP-NETs can be distinguished that differ in prognosis. Introducing this analysis into routine diagnostics could greatly improve therapeutic decision-making [24].

The current laboratory markers used in the diagnosis of patients with GEP-NETs are far from ideal and have various limitations. Numerous studies are underway in search of methods with the highest utility for the diagnosis and monitoring of patients with this disease. The use of appropriate laboratory analyses will reduce the use of imaging techniques, which are associated with high costs and increased harm to the patient. NETest, analysis of *13 HOX* and *HOTAIR* gene expression and evaluation of epigenetic modifications of the genome represent promising opportunities to modernize the laboratory diagnostic strategies used for patients with NETs.

### 2.7. Histopathological Findings

Holmager et al. leaned on the idea of using a Ki-67 index to assess the prognosis in patients with GEP-NETs [27]. In simple terms, a Ki-67 protein is a marker of cell proliferation. Its expression changes depending on the stage of the cell cycle [28]. The authors performed a retrospective observational study in which they assessed Ki-67-index. It was carried out at diagnosis and when disease progression was demonstrated. Material for the study had to be collected intraoperatively or by biopsy. The study showed that an increase in Ki-67 index correlates with disease progression. Moreover, both a high value of the index and its significant increase with radiologically demonstrated progression were associated with an increase in all-cause mortality. This result suggests that a Ki-67 index assay may be a good method for assessing prognosis. The authors also investigated if there was a difference in Ki-67 index for primary or metastatic cancers. The work showed that the index reaches similar values for both types of lesions [27].

## 3. Imaging Tests

### 3.1. Morphological Imaging

One of the key components of GEP-NEN patient diagnosis is diagnostic imaging. US, CT, MRI and nuclear medicine are the most useful ones. Recently, the development of nuclear medicine has become a very promising field, but CT is still the primary imaging modality for NEN patient diagnosis [29].

US has a relatively high sensitivity for detecting GEP-NEN liver metastases (85–90%), but a much lower sensitivity for detecting NENs in the pancreas (13–27%). However, it is possible to improve the sensitivity of this test by using contrast-enhanced US (CEUS). Another variation of US is endoscopic US (EUS) [30]. The advantages of EUS over US include the absence of interference from gases and subcutaneous tissue during the examination [10] (Table 3). It has a sensitivity of 54–97% for detecting pancreatic NETs (PanNETs), including a 71–94% sensitivity for detecting insulinomas [30]. Some studies indicate higher sensitivity and specificity in detecting PanNENs with EUS than with CT [10]. Moreover, a 25% increase in the detection of PanNENs was observed when EUS was used after CT. For these reasons, EUS is recommended as a complementary procedure to CT in suspected PanNENs. Furthermore, EUS has also been used in the diagnosis of gastric, duodenal and colorectal NENs. Because of the strong dependence of EUS efficacy in detecting NENs on their diameter, EUS is recommended in these structures only when the diameter is above 1 cm for gastric NENs and above 5 mm for rectal NENs. On the other hand, in the diagnosis of PanNENs, EUS also works well for smaller lesions [31]. In addition, the use of EUS provides the opportunity to perform EUS—fine needle aspiration (EUS-FNA). This method enables histological and cytological evaluations of the lesion and influences the decision regarding the applied treatment [10,31]. EUS-FNA plays a special role in the diagnosis of PanNENs, where this method allows us to distinguish lesions or neoplasms, with EUS patterns mimicking pancreatic NENs and the grading of PanNENs [31].

CT is very commonly used in the detection of NENs. Due to contrast, in addition to tumor detection, CT also provides accurate information about tumor location in relation to blood vessels and bile ducts. Based on the observed degree of necrolysis, tumor size and tumor infiltration, it is also possible to determine whether a given NET belongs to the G1/G2 or G3 classification. Contrast CT can also be administered via the enteral route when using CT enteroclysis/enterography. The methods are used to more efficiently evaluate NETs in the small intestine [30] (Table 3).

MRI and CT show similar sensitivities in the initial evaluation of the patient [32]. However, there are recommendations for the use of each modality. MRI appears to be superior in the evaluation of NENs in the liver, pancreas, rectum, bones and brain, while CT is recommended in the evaluation of NENs in the lung and small intestine and in the evaluation of vascularization [29,30,32]. MRI variations include diffusion-weighted imaging (DWI), intravoxel incoherent motion and diffusion kurtosis imaging. Intravoxel incoherent motion has been used to distinguish benign from malignant lesions, and importantly, does not require contrast. Diffusion kurtosis imaging allows for easier differentiation of NENs from other tumors in the head of the pancreas as well as benign from malignant lesions. Fortunately, changes in liver parenchyma such as fibrosis do not overly affect imaging results. Unfortunately, the imaging duration required is lengthy [30] (Table 3).

### 3.2. Nuclear Medicine

Nuclear medicine is a highly developed field used for imaging diagnostics and treatment strategies. This method works by a combination of a gamma- (SPECT)/positron- (PET)-emitting radionuclide, a bifunctional chelating agent (cyclic/acyclic), a linker and a biomolecule (receptor-specific agonist/antagonist) [33]. In the past, ^111^In-pentetreotide SPECT/CT was frequently used. However, numerous studies now urge its replacement with ^68^Ga-DOTA-SSA PET/CT, as this method achieves better diagnostic sensitivity and exposes the patient to less radiation [29,32,33,34,35]. ^111^In-pentetreotide SPECT is still used; however, studies have recently been performed to optimize the parameters involved in this process. Based on clinical and phantom studies, a statistically significant improvement in the contrast-to-noise ratio was observed with scatter correction (triple energy window) and attenuation correction at energy window widths of 171 keV ± 10% and 245 keV ± 7.5% and with ordered-subset expectation maximization by the use of 8 subsets and 6 iterations. The improvement of the contrast-to-noise ratio is intended to sharpen the contrast and boundaries of organs and lesions. These parameters are recommended for use in the clinic [36].

The currently recommended method for the detection of GEP-NETs is ^68^Ga-DOTA-SSA PET/CT [29,33]. This method shows greater sensitivity for detecting GEP-NETs than that achieved with standard imaging or imaging with ^111^In-pentetreotide SPECT [32,34,35]. The percentage of lesions detected in patients with GEP-NETs by ^68^GA-DOTATATE PET/CT and by ^111^In-pentetreotide and standard imaging (CT and/or MRI) were 95.1%, 30.9% and 45.3%, respectively. For 1/3 of patients, this study led to a change in treatment. This imaging method is also superior to ^18^F-FDOPA PET/CT at detecting both low-grade and high-grade GEP-NETs [35]. The advantages of GEP-NET diagnosis using ^68^Ga-DOTA-SSA PET/CT include increased sensitivity for detecting primary GEP-NETs and liver, bone and lymph node metastases. This method has also been shown to have a significant impact on treatment alterations in a large proportion of GEP-NET patients and on their overall survival and prognosis [32,34]. Furthermore, using ^68^Ga-DOTA-SSA PET/CT, it is possible to control the efficacy of PRRT and the use of chelators common to PET, and beta emitters (DOTATATE, DOTANOC, DOTATOC) enables precise PRRT [32,35]. These chelators, despite their differences in affinity for individual SSTRs, have similarly high affinities for SSTR2, making them effective at similar levels [34]. Recently, the use of new TRAP, THP and FSC chelators for ^68^Ga has been proposed to allow for the formation of more stable structures with radionuclides and, thus, to improve their use in diagnostics. The potential of using ^86^Yttrium or ^89^Zirconium as alternatives to ^68^Ga in centers without access to it has also been suggested [33].

^18^F-FDG PET/CT was initially recommended only for the detection of GEP-NETs with Ki-67 greater than or equal to 10% and a low expression of SSTRs, or of G3, NETs and NEC. Positive results of this test are considered a negative prognostic factor for patient overall survival [29,32,35]. Two new classifications of GEP-NENs different from those currently outlined by the WHO and based on ^18^F-FDG PET/CT and ^68^Ga-DOTATATE PET/CT have been recently proposed [37,38]. The recently published classification is based on three levels, C1–C3, where C1 is present when all ^68^Ga-positive lesions are ^18^F -, C2 is present when all lesions are ^68^Ga-positive and at least one of them is ^18^F+, and C3 is present when there is at least one ^68^Ga- and, at the same time, a ^18^F+ lesion [38]. Importantly, differences were noted in median progression-free survival (mPFS) across each group (*p* < 0.05) and in median overall survival (mOS) (*p* < 0.05, except for C1/C2 where *p* = 0.08). These are significantly better results in predicting mOS and mPFS than those achieved using the grades G1–G3 defined in the classification suggested by the WHO [38].

^18^F-FDG PET/CT has also been used as a prognostic factor for patients undergoing PRRT. There was a statistically significant relationship (*p* = 0.033) between the median overall survival of patients with GEP-NETs and whether they were ^18^F-FDG PET/CT positive (greater median overall survival) or negative after the second course of ^177^Lu-DOTATATE [39]. Additionally, a meta-analysis examining the prognostic values of ^18^F-FDG PET/CT in patients treated with PRRT with ^177^Lu and/or ^90^Y showed similar relationships. ^18^F-negative patients showed greater OS and PFS (*p* < 0.001 and *p* < 0.002) [40].

The newly used radionuclide is ^64^Cu. ^64^Cu-DOTATATE PET/CT was found to have a sensitivity of 100% and a specificity of 96.8% for finding NETs in patients [41]. In another experiment, the performance of ^64^Cu-DOTATATE PET/CT was compared with that of ^68^Ga-DOTATOC PET/CT. These tests appeared to have the same sensitivity for identifying NETs in patients, but ^64^Cu-DOTATATE PET/CT enabled the detection of significantly more NET lesions in the body than that achieved with ^68^Ga-DOTATOC PET/CT [42]. Furthermore, the use of ^64^Cu carries additional benefits relative to ^68^Ga use, such as easier study logistics as a result of a longer half-life of 12.7 h compared to the half-life of 1.1 h for ^68^Ga and with easier production. This assay is safe for patients [41,42].

An SSTR antagonist was also used in the diagnosis of patients with GEP-NETs. Studies using SSTR antagonists with radiolabeling were found to produce better imaging results than those observed using SSTR agonists with radiolabeling [34]. A study comparing the sensitivity of ^68^Ga-OPS202 and ^68^Ga-DOTATOC PET/CT for the detection of malignant lesions and liver lesions (including metastases) in patients with gastroenteropancreatic neuroendocrine tumors proved the superiority of the antagonist used (OPS202) over the SSTR agonist. The sensitivity of the former was 88–94% vs. 59% for the latter, *p* < 0.001. Furthermore, the method using ^68^Ga-OPS202 produced a better contrast between lesions and healthy tissue. This finding is due to the significantly lower SUVmax of healthy liver tissue for ^68^Ga-OPS202 than for ^68^Ga-DOTATOC [43]. A similar comparison was also performed with ^18^FAIF-NOTA-JR11 and ^68^Ga-DOTATATE. The ^18^F-NOTA-JR11-based assay produced a higher contrast between healthy tissue and NEN-induced liver lesions. This result may have been one of the reasons for the detection of more primary lesions and metastases of NENs in the gastrointestinal tract using ^18^F-NOTA-JR11 than with ^68^Ga-DOTATATE [44].

Insulinomas pose a significant diagnostic problem. In insulinomas, the expression of SSTRs is often insufficient for effective imaging with ^68^Ga-DOTA, and MRI is often ineffective (^68^Ga-DOTATAC PET/CT sensitivity 31–90%, MRI 68%) [34,35]. Therefore, PET imaging of GLP1R receptors is recommended here [29,32]. Based on this method, a sensitivity for the detection of insulinomas of 97.7% with ^68^Ga-NOTA-exendin-4 PET/CT and 94.6% with ^68^Ga-DOTA-exendin-4 PET/CT was achieved in two different studies [35,45]. In addition, ^68^Ga-DOTA-exendin-4 PET/CT was superior to ^111^In-DOTA-exendin-4 SPECT/CT and MRI in detecting insulinomas [45].

## 4. Systemic Treatment

Among patients with metastatic/unresectable NETs, systemic therapy is the treatment of choice. Depending on the histological differentiation of the tumor, the accepted treatment regimens are different. In the treatment of G1 and G2 stages of GEP NET SSA, targeted therapy, PRRT or chemotherapy are used. Additionally important is the origin of the neuroendocrine tumor, whether it is a pancreatic NET or midgut NET. For hepatic dominant lesions, chemoembolization or chemotherapy is the accepted form of treatment. Not all patients with metastatic or unresectable NETs require systemic therapy. If there are no symptoms of disease and no progression, then an observation and assessment of the progression with tumor markers is recommended. A separate group included patients with G3 NETs or NECs who also received treatment, as shown in Table 4.

Due to intensive research, there is a continuous development of the mentioned therapeutic methods. Advances that have been made recently are described below.

### 4.1. Somatostatin Analogs

Somatostatin analogs (octreotide LAR and lanreotide) can be used in patients with hormonally active tumors in which receptors for the somatostatin SSTR have been demonstrated. Therefore, it is important to perform SSTR imaging before initiating therapy [46,47]. SSAs are mainly used to control the disease symptoms occurring due to the overproduction of the somatostatin hormone by neuroendocrine cells.

The SSAs in use are administered intramuscularly (octreotide LAR) and subcutaneously (lanreotide autogel). We are currently recruiting for a study to evaluate the efficacy and safety of a subcutaneously administered depot formulation of octreotide (CAM2029) in patients with advanced, well-differentiated GEP NET [NCT0505094]. The study would be designed to compare the results obtained using the new form of drug administration with those of octreotide LAR and lanreotide autogel, and to confirm that the new formulation would provide a significant convenience for patient use.

Due to two randomized controlled phase III trials (PROMID and CLARINET), the antiproliferative effect of somatostatin analogs in anticancer treatment has been proven. A significant prolongation of progression-free survival (PFS) was observed. The PROMID study used octreotide LAR on a group of patients with advanced NETs located in the midgut with a Ki-67 score ≤ 2% in 95% of cases and with liver metastases ≤ 10% in 75% of patients. There was a significantly higher median time to PFS in treated patients than in placebo-treated patients (*n* = 85, 14.3 months vs. 6.0 months; risk factor (HR), 0.34; 95% confidence interval (CI), 0.20 to 0.59) [48]. The CLARINET trial, in which lanreotide autogel was used, was conducted among patients with advanced GEP-NETs or NETs of unknown origin, with Ki-67 indices ≤ 10% and hepatic tumor involvement of ≤ 25% in 67% of patients. There were results of prolonged median PFS in the treated group compared to that in the placebo group (*n* = 204, 32.8 months, vs. 18 months, HR 0.47; 95% CI, 0.30 to 0.73) [49]. In view of the above results and the good tolerability of SSA, octreotide LAR and lanreotide autogel are recommended for the first-line treatment of unresectable or metastatic high- or intermediate-grade (G1, G2) GEP-NETs with low to intermediate tumor volumes (Ki-67 < 10%).

A possible therapeutic option for a minor response would be to increase the dose of SSA. The CLARINET FORTE study, a phase II study in patients who had progressed on standard-dose lanreotide (once every 28 days), tested the effectiveness of increasing the dose of SSA (once every 14 days) and its association with dose-related toxicities. PFS was shown to extend to 5.6 months for patients with panNETs and 8.3 months for patients with midgut NETs [50]. The side effects observed were typical for SSA: nausea, abdominal pain, fatty diarrhea, gas, hyperglycemia and gallstones. No toxic effects were observed [51].

### 4.2. INF Alpha

Interferon alpha (INF-α) has antiproliferative and antimitotic properties and promotes the influx of immune cells. Similar to SSA, it has antisecretory properties [52], but due to its higher number of side effects, it is now a rarely used drug.

The Southwestern Oncology Group conducted a prospective, randomized phase III study to evaluate the efficacy and safety of depot octreotide in combination with INF alpha-2b compared to those of depot octreotide with bevacizumab in advanced carcinoid syndrome, which can occur in neuroendocrine tumors. The median PFS values were 15.4 and 16.6 months, demonstrating comparable results [53]. Therapy for carcinoid syndrome was also addressed in the phase III TELESTAR trial evaluating the safety and efficacy of ethyl telotristat, a tryptophan hydroxylase inhibitor that blocks the early serotonin pathway. The study was conducted on a group of patients taking SSA with carcinoid syndrome. Treatment efficacy was higher in the ethyl telotristat group than in the placebo group (PFS 54.0% (250 mg) and PFS 89.7% (500 mg) for ethyl telotristat), confirming the safety and efficacy of the drug. These results are also significantly higher than those obtained in studies with INF alpha, which may lead to the replacement of INF alpha with a new drug in the future [54].

### 4.3. Radionuclide Treatment

Peptide receptor radionuclide therapy (PRRT) is a therapeutic option that is developing in a promising direction. Based on the results of the NETTER-1 trial [55,56], PRRT has been approved for highly differentiated, G1 G2 advanced, unresectable GEP NETs with metastasis that express SSTR [57]. NETTER-1 was a randomized, controlled trial comparing the use of ^177^Lu-DOTATATE in combination with LAR octreotide (30 mg every 4 weeks) with the use of higher doses of LAR octreotide (60 mg every 4 weeks) in patients with advanced G1 G2 midgut GEP NETs with SSTR expression who progressed on SSA therapy. The results demonstrated higher PFS and RR in the group receiving ^177^Lu-DOTATATE in combination with octreotide LAR than those of the group treated with octreotide LAR alone (*n* = 229; 20-month PFS 65.4% vs. 10.8%; RR 18% versus 3%; HR, 0.21, 95% CI, 0.13 to 0.33) [56].

Because the NETTER-1 trial included only a group of patients with midgut NETs, studies are currently underway to investigate the use of PRRT in patients with pancreatic NETs and non-midgut NETs. The randomized phase III COMPETE trial [NCT03049189] for inoperable, progressive G1, G2 GEP NETs with SSTR expression is comparing the use of ^177^Lu-EDOTREOTIDE with the use of everolimus. The NETTER-2 trial [NCT03972488] is a phase III study evaluating the efficacy of ^177^Lu-DOTATATE in combination with SSA LAR compared to that achieved with high doses of SSA, which is used as first-line therapy in patients with G2 and G3 GEP NETs with SSTR expression who have a Ki-67 index of 10–55%. 

Despite promising treatment prospects, PRRT directed against tumor cells is not inert to other organs and can lead to toxic damage, mainly to the liver, kidney and bone marrow. In the NETTER-1 trial listed above, adverse effects that have been reported include myelosuppression (leukopenia, anemia, thrombocytopenia), renal failure and carcinoid syndrome [55,58]. Contraindications to ^177^Lu-DOTATATE use in patients with GEP NETs include GFR < 30 and liver failure [59]. The limitations posed by the therapy are also due to the heterogeneous expression of serotonin receptors and, thus, the partial resistance of tumor cells to the therapy [60]. Therefore, ^177^Lu-DOTATATE is co-administered with amino acid infusion, usually arginine or lysine, to prevent nephrotoxicity [61]. Intrahepatic-arterial (IAH) infusion of radio-labeled somatostatin analogs is being evaluated in the currently ongoing LUTARTERIAL trial [NCT04837885] among patients with GEP NETs who predominantly exhibit liver metastases. This route of administration could be a more promising option than standard intravenous delivery. Liver-selective internal radiation therapy (SIRT) using a 90 Yttrium microsphere scan is another possible option. It is being considered in patients with GEP NETs with liver metastases and liver failure and in patients who have contraindications to chemoembolization [62].

The field of research on new peptides and PRRT therapies is expanding to improve efficacy and safety. ^177^Lu-DOTA-JR11 is an SSRT antagonist, and its mechanism of action has been shown to also involve its attachment to inactive SSTRs within the cell membrane, resulting in increased uptake of radionuclide substances and increased tumor irradiation [63]. Fibroblast activation protein inhibitors (FAPIs) are a new group of drugs used for anticancer radiotherapy. Many cancers, including NETs, have a high tolerance for FAPI treatment [64]. In a preclinical study, novel ^177^Lu carriers such as EB-TATE were considered. The results suggested that the binding of radioisotopes to albumin could prolong the half-life of the drug, improve the tumor/kidney dose ratio and improve the regression rate of GEP NETs [65,66]. A phase I trial using ^177^Lu-EB-TATE in patients with NETs is currently underway to evaluate the safety of this therapy [NCT03478358].

PRRT is also being considered as a new addition to other therapeutic modalities. The effect of the addition of PRRT before or after surgical resection has been studied in patients with pancreatic NETs. It has been proven that the use of PRRT before surgery is associated with higher PFS than that achieved with the use of PRRT after surgery [67]. Currently, there is another ongoing study on the use of ^177^Lu-DOTATATE before or after surgical removal among patients with SSTR-positive GEP NETs [NCT04609592]. The effects of combination treatment with chemotherapy are also being evaluated. The CONTROL NETS trial is an ongoing randomized trial in which the use of the combination of cabecitabine with temozolomide (CAPTEM) plus ^177^Lu-DOTATATE is compared with the use of CAPTEM or ^177^Lu-DOTATATE as a monotherapy [68]. The combination of Lu-177-DOTATATE with olaparib in unresectable GEP NETs is also under evaluation [NCT04086485]. We report the results of a phase I/II study of the combination of everolimus with ^177^Lu-edotreotide among patients with GEP NETs and lung NETs. The mean ORR response rate of the combination therapy was only 9%. PFS in the study group was 23.3 months, and adverse effects observed in 36% of patients were grade 3 toxicities and included conditions such as pneumonia, fatigue, and neutropenia [69]. Currently, there is also a study evaluating the efficacy and safety of PRRT with ^177^Lu-edotreotide as the first or second line of treatment in comparison to the accepted best standard of care—CAPTEM (capecitabine- temozolomide), everolimus, and FOLFOX (folic acid + 5-fluorouracil + oxaliplatin). The study involves a group of patients with highly differentiated G2 and G3 NETs originating from the pancreas with preserved SSTR expression [NCT04919226].

### 4.4. Transarterial Treatment

In the case of G1 and G2 stage GEP NETs with predominant hepatic lesions or with the presence of liver metastases, chemoembolization is the accepted form of treatment. Since the liver lesions are highly vascularized by branches of the hepatic artery, it is possible to identify the vessel supplying the tumor and embolize it. The modern technique of transarterial mechanical embolization (TAME) is applicable here. Among TAMEs, chemoembolization (TACE) is currently considered the most important. It is based on the administration of a chemotherapeutic agent directly into the vessel [70]. Touloupas et al. [71] conducted a cohort study in which they confirmed the long-term efficacy of TACE. A study is also currently underway to compare TACE with bland embolization (BE) and with embolization by drug-eluting beads (DEB) in terms of hepatic progression-free survival (HPFS). In addition, the symptom-relief interval and the harmfulness of each method will also be evaluated [NCT02724540]. For radioembolization (TARE), analogously, high doses of radiation are delivered via the vascular route to the tumor. At this point, the method is not yet part of the treatment regimen but rather is still in the research stage [70]. However, this approach offers high hopes for its development as well as its introduction into routine use.

### 4.5. Targeted Therapy

Targeted therapy, which currently uses two drugs, everolimus and sunitinib, is used to treat nonsurgical GEP NETs with metastasis [72]. Everolimus, a selective inhibitor of the mTOR (mammalian target of rapamycin) pathway, is recommended in patients with advanced-stage G1 and G2 GEP NETs. Its efficacy has been evaluated in two randomized phase III trials [73]. The RADIANT-3 trial, where everolimus was used in patients with PanNETs, achieved a significantly higher median PFS for treated patients than for placebo-treated patients (*n* = 407; PFS 11.0 months vs. 4.6 months; HR, 0.35; 95% CI, 0.27 to 0.45) [74]. In the RADIANT-4 trial, patients with progressive, endocrine-inactive NET tumors of the lung or gastrointestinal tract were randomly assigned. Additionally, treated patients had satisfactorily higher PFS than that of those treated with placebo (*n* = 302; PFS 11.0 months vs. 3.9 months; HR, 0.48; 95% CI, 0.35–0.67) [75]. The side effects that were observed were gastritis, rash, diarrhea, fatigue, anemia, and antihyperglycemic effects.

Sunitinib is a multiselective tyrosine kinase inhibitor (TKI) that belongs to the angiogenesis inhibitor group. It was evaluated in a randomized phase III trial where patients with pancreatic NETs were considered, and the primary blanket mean time to PFS was 11.4 months in the treated group vs. 5.5 months in the placebo group (*n* = 171; ORS 9.3% vs. 0% in the placebo group; HR, 0.42; 95% CI, 0.26–0.66). The side effects that have been observed with sunitinib include nausea, vomiting, diarrhea, fatigue, and chronic heart failure. Sunitinib is recommended for use in patients with PanNETs in the advanced stages G1 and G2 [76].

Due to the approval of sunitinib only in the strict indication of panNETs, studies are currently underway to investigate the use of other angiogenesis inhibitors in the treatment of extrapancreatic NET tumors as well [77]. The ongoing SANET-p phase III, randomized, placebo-controlled trial evaluating surufatinib in highly differentiated PanNETs showed a favorable PFS (*n* = 172; PFS 10.9 months vs. 3.7 months; ORR 14%). The observed side effects that occurred were hypertension, proteinuria, and hypertriglyceridemia [77]. The SANET-ep phase III randomized trial in patients with advanced, highly differentiated epNETs showed a promising PFS of 9.2 months with surufatinib treatment versus 3.8 months in the placebo group (*n* = 289; ORR 10%) [78]. A parallel phase I/II study conducted in the United States also showed promising PFS with surufatinib in patients with PanNETs and epNETs previously treated with everolimus or sunitinib [79]. The positive results of these studies led to the FDA approval of surufatinib for the treatment of patients with advanced, progressive, highly differentiated NETs of pancreatic and extrapancreatic origin.

Within tyrosine kinase inhibitors, in addition to sunitinib, many new drugs are being evaluated in prospective studies for their possible use in patients with advanced GEP-NETs. The need for research in this field is also greater in view of the satisfactory but limited clinical efficacies of everolimus and sunitinib in the treatment of patients with NENs due to the development of cellular resistance mechanisms and downregulation of the number of receptors targeted by therapy [80]. There is an ongoing phase III CABINET clinical trial of cabozantinib, a novel tyrosine kinase inhibitor, in patients with GEP NETs and alveolar NETs who have progressed on at least one line of treatment other than SSAs [NCT03375320].

Anti-angiogenic agents, due to their affinity for vascular endothelial growth factor (VEGF) receptors, are now a focus for the development of targeted therapies for NETs. Another drug currently under research is bevacizumab, a humanized monoclonal anti-VEGF antibody [81]. Preclinical studies suggest that bevacizumab can induce an immune response by modulating the tumor immune microenvironment. It may increase the ratio of antitumor immune cells to protumor immune cells [82]. Bevacizumab is currently being investigated in combination with folic acid/5-fluorouracil/anginothecan as a second-line treatment in NECs [NCT02820857].

The efficacy of everolimus in combined therapies is also currently being evaluated. Due to the recent approval of everolimus for use in advanced PanNETs, it is necessary to determine the best treatment sequence, including the use of previously used agents. The ongoing SEQTOR trial compares the use of everolimus before or after palliative chemotherapy in the form of STZ-5FU [NCT02246127]. There are also ongoing studies that evaluate the efficacy of everolimus compared to that of PRRT [NCT03049189].

### 4.6. Immunotherapy

The emergence of new therapeutic approaches, such as checkpoint inhibitors (CPIs) targeting the programmed death protein PD-1 or the cytotoxic T lymphocyte-associated protein 4 CTLA-4, has opened completely new possibilities for the treatment of various tumors [24]. Despite extremely promising effects against other cancer types, checkpoint inhibitors have thus far shown limited efficacy in the treatment of patients with NETs [83,84,85]. Immunotherapy with single CPI agents has not shown significant antitumor effects in patients with highly differentiated GEP NETs, except among patients with microsatellite-unstable tumors [86]. Additionally, for patients with highly differentiated NETs and an elevated Ki-67 index, there is a chance of a better response to treatment. This finding is due to the significant role of the microenvironment surrounding the tumor, mainly the influx of lymphocytes within the tumor, PD-1 expression levels, and microsatellite instability (MSI). Typically, these factors are associated with advanced forms of NETs and poorer prognosis [87,88,89].

The PD-1 protein is an important point of capture in immuno-oncology. The interaction of PD-L1 found on the surface of tumor cells with PD-1 on the surface of T lymphocytes leads to immune suppression within the tumor. The application of a PD-L1 inhibitor can interrupt suppression and affect the immune response. The phase II KEYNOTE-158 study evaluated the activity of premolizumab, a monoclonal antibody against the programmed cell death protein PD-1, in a group of patients with highly differentiated NETs (77.5% GEP NETs). The endpoint of the study was the percentage of objective ORR responses. The ORR was 3.7% (all partial responses), and responses occurred in patients without PD-L1 expression. Seventy-five percent of the subjects had a Ki-67 index >10%. The median PFS in the study group was 4.1 months [90]. Another anti-PD-1 antibody whose activity was evaluated, was spartalizumab. In a phase II study of 116 patients (56% with highly differentiated GEP NETs), only one patient with GEP NETs showed a partial response. No correlation was observed between the degree of response and PD-1 expression [39]. Additionally, in 2021, a phase II study was completed to evaluate the efficacy and safety of spartalizumab (PDR001) in patients with advanced-stage nonsurgical or metastatic GEP NETs who had progressed during prior treatment. The ORR was 7.4% in this group of subjects [NCT02955069]. A randomized phase III trial of cabozantinib for the treatment of patients with NETs and advanced carcinoid syndrome is currently underway. Cabozantinib, which is a specific tyrosine kinase inhibitor, can slow tumor growth if therapy is effective [NCT03375320]. Due to the mediocre results of studies of individually tested therapies, the trials of combined therapies seem to be promising. One such evaluation was the phase II DART study, in which the use of combined therapy of anti-PD-1 with anti-CTLA-4 (nivolumab with ipilimumab) was evaluated among patients with epNET stages G1/G2 and G3. The results proved to be promising, with an RR of 44% for G3 patients compared with an RR of 0 for G1/2 patients. This finding thus supports the possibility of using combined immunotherapy in patients with advanced stages of NETs [91].

### 4.7. CAR-T-Cell Therapy

Great hopes are also placed on the intensely developing field of immunotherapy, which includes methods such as tumor-infiltrating lymphocytes (TILs), T-cell receptors (TCRs) and chimeric antigen receptors (CARs) [92]. CAR- T cells are genetically recombinant T cells consisting of an antigen recognition domain and an intracellular signaling domain that are designed to recognize specific tumor cells and induce their apoptosis. CAR-T therapy targeting the CD-19 antigen has been a breakthrough in the treatment of B-cell acute lymphoblastic leukemia. Due to the disproportionately high treatment efficacy of this approach compared to that achieved using other methods in this context as reflected by a probability of survival for at least 1 year of >90%, CAR-T therapy is currently under intensive investigation in patients with NENs [93].

The aim of the researchers was to construct CAR- T cells directed against SSTR and characterized by effectiveness and safety in the treatment of patients with GEP NETs. In a preclinical experimental study, T cells modified by the addition of octreotide molecules and CD28 were transfected into PanNET (BON1 and QGP1) lung NET cell lines and NECs (HROC-57 and NEC-DUE1) in mouse models. Presented at the North American Neuroendocrine Tumor Society symposium in 2019, the results showed that the modified cells localized to tumor cells and significantly inhibited their growth in BON1 cell lines. Additionally, CAR-T cells had higher expression levels of IFN alpha and IFN gamma than those of normal T cells, indicative of additional immune system activation [94]. The next step will be a clinical trial.

### 4.8. Bispecific Antibodies

Bispecific antibodies (BsAbs) also seem to be interesting. In an ongoing phase I study currently undergoing recruitment, the XENCOR bispecific antibody SSTR2/CD3 (XmAb18087) is being evaluated. Its mechanism of action involves bilateral binding of receptors—first, with the receptor for somatostatin analogs STR2 on tumor cells and second, with cluster of differentiation (CD) 3—occurring on the surface of cytotoxic T lymphocytes. This process causes an influx of immune cells into the NET area and an increased inflammatory response. We are studying patients with G1 G2 PanNETs, epNETs and lung NETs after progression on an SSA and one other targeted agent [NCT03411915]. In the phase I/II study, the use of pentarin PEN-221, which is a miniature antibody–drug conjugate that binds to maytansinoid emtansine (DM1)- and SSTR 2-expressing tumors, is being evaluated [95]. Phase I results were presented to the American Society of Clinical Oncology in 2018. No patients experienced an ORR; however, three subjects experienced a minor response. The results were for patients previously treated with PRRT for small-cell lung cancer who were unresponsive, but recruitment is already underway for a group with small intestinal and pancreatic NETs.

### 4.9. Vaccines

Oncolytic viruses produced by genetic engineering are the future of immunotherapy. They are able to bind to selective molecules and induce an immune response against tumor cells. In one study, the researchers constructed an oncolytic virus expressing a PD-L1 inhibitor and granulocyte-macrophage colony-stimulating factor (GM-CSF). The secreted PD-L1 inhibitor was shown to bind to PD-L1 on tumor cells, which blocks its binding to PD-1 on the surface of T cells, promotes the bypassing of tumor cell immunosuppression and enables the stimulation of the antitumor immune response. GM-SCF expression further contributes to the influx of immune cells. However, the use of PD-L1 inhibitors may lead to the development of tumor cell resistance to therapy as a result of upregulation of PD-L1 expression. Further research on oncolytic vaccines is necessary [96]. The oncolytic adenovirus (AdVince) is engineered to affect the selective replication of human chromogranin An in the tumor area and to promote homing to NET cell growth. In preclinical phase I/II studies, the therapeutic premise was confirmed; adenovirus caused an increase in chromogranin A replication equally without affecting the increase in the influx of other inflammatory cells and led to apoptosis of NET cells [97] [NCT02749331]. We also report a significant antitumor response in NET/NEC using an oncolytic vaccine containing GLV-1 h68 virus. Preclinical studies were conducted on panNET (BON-1, QGP-1), lung NET (H727, UMC-11) and NEC (HROC-57, NEC-DUE1) cell lines. In addition, it has been shown that the use of the vaccine does not adversely affect the efficacy of everolimus, offering the possibility of future combination therapy [98]. Another oncolytic vaccine recently tested contains herpes simplex virus HSV-1. The results indicate that the virus inhibits cell proliferation in G3 NECs in a mouse model. Conducting further clinical trials may lead to the introduction of this vaccine as a form of immunotherapy for patients with NETs [99].

### 4.10. Cytotoxic Chemotherapy of Highly Differentiated G1 and G2 GEP NETs

Chemotherapy is not a common treatment option in patients with G1 and G2 highly differentiated tumors; however, it is used in the treatment of patients with unresectable highly differentiated metastatic panNETs. With Ki-67 index values >10%, in hormonally active panNETs, it is the treatment of first choice (Table 1).

Alkylating drugs used in combination with the antimetabolite streptozocine (STZ) in combination with 5-fluorouracil (5-FU) or temozolomide used in combination with capecitabine (CAPTEM) are used for the above indications [100]. Decarbazine or doxorubicin are also used in progressive disease [101]. However, due to their short half-lives and significant toxicities, alkylating drugs such as streptozocin (STZ) are now rarely used [102]. The ongoing SEQTOR trial [NCT02246127] is evaluating the safety and efficacy of STZ/5-FU therapy in combination with everolimus in patients with PanNETs.

However, CAPTEM therapy has received the most attention. A past retrospective study showed that temozolomide in combination with capecitabine achieves an objective response rate (ORR) of 70% and a median PFS of 18 months among patients with panNETs [103]. More recent results from an ongoing randomized phase II trial [NCT01824875] also comparing CAP/TEM with CAP or TEM monotherapy among patients with progressive unresectable G1 G2 panNETs are equally promising [104]. Patients receiving CAP/TEM therapy have been demonstrated to have higher a ORR and PFS than those of patients receiving monotherapy (*n* = 144, PFS 22.7 months vs. 14.4 months; HR 0.58; ORR 33.3%) [9,105]. Despite a noticeably lower ORR than that observed in the original study, CAP/TEP therapy is recommended for the treatment of unresectable, progressive PanNETs in stage G1 G2 due to its efficacy.

For the treatment of extrapancreatic neuroendocrine tumors, chemotherapy in the form of 5-FU, capecitabine and oxaliplatin may be considered if other therapies fail or disease progression occurs [106].

### 4.11. Cytotoxic Chemotherapy of Low-Differentiated G3 GEP NETs and GEP NECs

In contrast to highly differentiated G1 G2 GEP NETs, cytotoxic chemotherapy is the gold standard for patients with advanced, progressive low-differentiated G3 unresectable, metastatic GEP NETs. Regrettably, there are no objective data on recommended therapies for patients with stage G3 GEP NETs [107,108]. Thus, the choice of a definitive drug remains a matter of debate.

According to the results of several studies, CAP/TEM appears to be effective among patients with G3 pan NETs with a Ki-67 index <55%. The ORR of the above studies ranges from 33% to 70% [109]. Additionally, a retrospective, multicenter study that compared TEM treatment among groups of patients with G3 PanNETs and G3 NENs showed a higher ORR among the G3 NET group [110]. The superiority of CAPTEM therapy was also confirmed by a study conducted by Nordic investigators, which showed that among a group of patients with Ki-67 < 55% G3 PanNETs, treatment with etoposide in combination with cisplatin is not effective [111]. On the other hand, in patients with Ki-67 >55% G3 PanNETs, chemotherapy with etoposide in combination with cisplatin is considered a first-line drug [107]. Treatment of patients with aggressive G3 NETs with platinum plus etoposide is equally controversial. There are studies demonstrating poor responses to this treatment [112,113]. Among potentially effective therapies for patients with G3 NETs are everolimus, sunitinib or PRRT, which are mentioned in the European Society tor Medical Oncology guidelines [1].

Regarding patients with GEP NECs, this group is so heterogeneous that they require a different treatment from that used in patients with GEP NETs. The choice of therapy depends primarily on the histological stage and, to a lesser extent, on the origin of the cancer. According to the results of the NORDIC study, it can be classified on the basis of the Ki-67 index with a cutoff point of 55% so that two populations can be distinguished among patients with GEP NECs [111]. In the group with a high Ki-67 proliferation rate of 80–90%, TP53 mutation often coexists, and chemosensitivity to treatment is high. The group associated with DAXX/ATRX mutation, on the other hand, has lower tumor aggressiveness but is also less sensitive to treatment. In general, G3 GEP NECs, due to their high degree of cell division, are more sensitive to cytotoxic treatment than highly differentiated G1 G2 tumors, but their prognosis is much worse. Among the available treatments for patients with G3 NECs, etoposide plus cisplatin is the most commonly used. In a multicenter, retrospective, randomized study in patients with G3 NETs and NECs, it was shown that etoposide combined with cisplatin in G3 NECs achieved a median PFS of 4 months, a median OS of 12 months, and RR of 31% [114,115]. Irinothecan in combination with 5-fluorouracil (FOLFIRI) is recommended as a second-line therapy for patients with NECs [116]. FOLFOX therapy (folic acid + 5-fluorouracil + oxaliplatin) is also possible [107]. Temozolomide is also mentioned as a possibly effective therapy for patients with G3 NECs [117]. Other sources report that irinotecan and oxaliplatin may be second-line drugs, but their efficacies are low [112,114]. Due to the unsatisfactory results of cytotoxic treatment, it is necessary to develop research on new therapeutic approaches. The combination of ipilimumab with nivolumab in patients with epNECs, which was developed in a recent phase II trial, showed promising results [91]. In addition, a prospective, multicenter, nonrandomized open label phase II study of platinum-doublet chemotherapy in combination with nivolumab among patients with GEP NENs of unknown origin is currently ongoing [NCT03980925].

## 5. Surgery

Surgery is one of the most important treatment for patients with GEP-NENs, especially at lower stages. Surgery with lymphadenectomy, particularly minimally invasive procedures, is considered the gold standard curative treatment in localized disease, but is also important in multimodal disease in patients with advanced GEP-NENs [4]. This is because the type of surgical treatment is one of the independent factors of overall survival [118].

Surgery for higher-grade lesions has been variously evaluated because of the potentially small degree of improvement for the patient and complications. Nevertheless, new studies show that surgery can provide therapeutic benefit to patients, not only, but mainly in patients with the possibility of complete excision of lesions [119]. Radical surgery may be regarded in selected cases of advanced GEP-NENs [120].

Palliative surgery, which aims to reduce functional or tumor mass-related symptoms, also plays a large role [121], although early mortality after surgery may be higher [118].

### 5.1. Stomach (Gastric NETs, G-NETs)

They are classified into 3 categories related to pathological type:Type I most commonly manifests as small multiple, low-grade tumors [32]; they account for 70–80% of G-NET patients, and are more frequent in females. It is histologically composed of enterochromaffin-like cells [122]. Small tumors should be surveilled or resected yearly, depending on the size and depth of invasion [32,123]. According to ENETS guidelines from 2016, resection is recommended for lesions equal or above 1 cm, and those for which infiltration of the mucosa propria is anticipated. If feasible, resection could be performed using EMR (endoscopic mucosal resection) or preferred ESD (endoscopic submucosal dissection). In case of positive margins or T2 stage, local excision or partial gastrectomy should be considered [123].Type II develops in MEN-1 syndrome, and its treatment depends on the presence and size of other coexisting NETs (mainly duodenal and pancreatic) [32,123].Type III is usually diagnosed in late lesions, which require partial or complete gastrectomy with lymph nodes dissection [123], while the treatment of tumors below 2 cm is controversial and may be considered in a narrow range of patients [62]. In small tumors, endoscopic resection is recommended [123].

### 5.2. Duodenum (D-NETs)

Patients with duodenal NETs, which are small and not periampullary tumors that are nonfunctioning and limited to the submucosal layer, qualify for conservative treatment, that is, endoscopic resection. Radical duodenectomy with lymphadenectomy should be considered to be performed for patients with larger or more aggressive lesions [32].

The ER of D-NETs is associated with a high risk of complications and difficulty in achieving radical resection [62].

### 5.3. Pancreas (PanNETs)

In patients with small (<2 cm, especially smaller than 1 cm [124]), nonfunctional NETs, active surveillance is recommended [125].

Pancreatic resection with lymphadenectomy is recommended for all lesions larger than 2 cm and those lesions that produce symptoms (functioning) or have radiographic features of aggressiveness. There is no consensus about the minimal number of harvested lymph nodes, which varies from 7 to 13 [32,126].

National Comprehensive Cancer Network (NCCN) recommend considering more aggressive resections in tumors bigger than 1 cm. Nevertheless, the proposed limited resection (enucleation) can have similar outcomes as a complete resection, so should be considered [127].

### 5.4. Small Intestine (SI-NETs, SB-NETs)

Incidental diagnosis is rare, and most of these NETs show features of aggressiveness. Because of this, SB-NETs should always be treated with intestinal resection with lymphadenectomy [128].

### 5.5. Appendix (Appendiceal NETs, A-NETs)

Most commonly, A-NETs are diagnosed incidentally after appendectomy following appendicitis [129]. Preoperative diagnosis is extremely rare, not least because the differential diagnosis from appendicitis is difficult. For all GEP-NEN lesions in the appendix, surgical removal is indicated [130]. Usually, appendectomy of a G1 lesion, ≤1 cm, with limited infiltration (<3 mm), located not at the base of the appendix and without features of invasion into the vessels, is considered complete [62]. There is no recommendation as to whether a simple appendectomy or an additional right hemicolectomy (RHC) is appropriate for G2 and <2 cm lesions [62]. RHC is indicated for tumors larger than 2 cm [131] or with atypical histology. Lesions between 1 and 2 cm should be evaluated for risk, and appropriate treatment should be selected.

It is worth noting that some studies show positive effects of radicalized management [132], and in some studies, the relevance of mesoappendix invasion to decision-making is questioned [129,132].

There is no need for catamnesis when complete tumor resection has been achieved and for patients after RHC with lymphadenectomy with tumors < 2 cm, low Ki-67, and no node metastases [32,62]. Surgical excision of lesions located in the appendix carries the lowest relative risk (compared to that of surgical treatment for other types of GEP-NENs) [133].

### 5.6. Colon (C-NETs)

Colonic NETs are usually high-grade and poorly differentiated and demand informed surveillance [32]. Therefore, it is now recommended for patients with localized C-NETs to always receive a colectomy with lymphadenectomy [134] (best approximately 12 lymph nodes [135]). Lymphadenectomy is important because it is one of the strongest prognostic factors [136].

### 5.7. Rectum (R-NETs)

Rectal NETs are usually small, low-grade tumors with a low risk of metastases [32]. Thus, most R-Nets without signs of aggressiveness can be excised during endoscopy [137], especially with advanced ER techniques [32,62]. Small tumors can also be resected by transanal endoscopic microsurgery (TEM), which has the advantage of avoiding segmental resection [138]. However, for patients with tumors larger than 1.5 cm and with features of aggressiveness, a low anterior resection with total mesorectal excision should be performed [137]. Follow-up should be based on endoscopic examinations and MRI of the region [32,62].

### 5.8. Liver

The liver is the organ most frequently affected by GEP-NEN metastases [70] but can also be the primary site of malignancy. Surgical treatment depends on the locations and number of segments or lobes taken. In diseases occurring in two adjacent segments or in one lobe, parenchyma-preserving liver resection with radiofrequency ablation (RFA) is recommended. Transarterial chemoembolization or radioembolization (TACE/TARE) may also be performed. However, in bilobar metastasis with dominance in one lobe, single-stage surgery with RFA, staged resection or mixed therapy is acceptable [121].

## 6. Radio Guided Surgery

Another area being developed is radioguided surgery. This technique is based on the detection of somatostatin analogs with radionuclide attached to GEP-NETs by a handheld probe. This device then provides auditory/visual feedback to the surgeon about the presence of the lesion, allowing for precise intraoperative detection and differentiation from healthy tissue. The techniques used in this method mainly involve ^111^Indium-Pentetreotide, Technetium-99 m and ^68^Ga-DOTA-peptides. The use of these methods has resulted in a favorable increase in sensitivity and in the number of GEP-NETs detected during the procedure and sometimes in a reduction in the duration of the procedure. However, work on this topic has numerous limitations. There is a lack of data on the impact of these methods on the overall survival of patients, and the patient groups studied were rather small [139].

## 7. Advances in Minimally Invasive Surgery

The method of choice for the surgical treatment of small intestinal NENs (SI-NENs) is open resection. Although laparoscopy is an effective and modern method, classical surgical techniques are preferred for the treatment of patients with SI-NENs. In the case of diagnosing multifocal disease, the ability to obtain tactile feedback plays a major role. Sometimes, during palpation of the small intestine, lesions are detected that have not been diagnosed by imaging. Laparoscopy does not allow for this. The risk of incomplete resection and insufficient training of surgeons to perform laparoscopy discourages its frequent use [140]. However, there are reports of the superiority of minimally invasive techniques in the area described. Kaçmaz et al. [141] compared the outcomes of surgical treatment of SI-NENs by open and laparoscopic methods. The results showed a significantly higher 5-year survival rate in patients treated with the minimally invasive method than in those treated with open resection. The values were 84% and 71%, respectively. Furthermore, it has been suggested that multifocal disease may not be a contraindication to laparoscopy, as recent studies have negated the correlation between multifocal disease and overall survival. This hypothesis needs to be confirmed but offers hope for expanding the indications for minimally invasive surgery. The results of the aforementioned study shed new light on the use of laparoscopy in the treatment of patients with SI-NENs [141].

## 8. Conclusions

With the increase in the incidence of GEP-NENs, there is an increased interest in improving patient diagnosis and treatment. The above work has demonstrated numerous possible options for these applications. The low specificity of the clinical manifestations of neuroendocrine tumors presents diagnostic difficulties; however, there are reports of success in the area of laboratory diagnostics. Nuclear medicine deserves special attention as it is a rapidly developing field in diagnostic imaging, radioisotope treatment and radioguided surgery. There is still much research underway that has the potential to provide additional data that will be helpful in supporting further innovations to the diagnosis and treatment of patients with GEP-NENs.

## Figures and Tables

**Table 1 cancers-14-02028-t001:** Classification and grading criteria for neuroendocrine neoplasms (NENs) of the gastrointestinal tract and hepatopancreatobiliary organs after the WHO 2019 classification.

Terminology	Differentiation	Grade	Mitotic Rate (mitoses/2 mm^2^)	Ki-67 Index
NET, G1	Well differentiated	Low	<2	<3%
NET, G2		Intermediate	2–20	3–20%
NET, G3		High	>20	>20%
NEC, small-cell type	Poorly differentiated	High	>20	>20%
NEC, large-cell			>20	>20%
MiNEN	Well or poorly	Variable	Variable	Variable

NEC: Neuroendocrine carcinoma; MiNEN: mixed neuroendocrine-nonneuroendocrine neoplasms; NET: neuroendocrine tumor; Ki-67 index: rate of cell growth.

**Table 2 cancers-14-02028-t002:** Selected genetic diseases associated with gastroenteropancreatic neuroendocrine neoplasms (GEP-NENs).

Illness/Phenotype	Pattern of Inheritance	Causative Gene(s)	Products of Genes Expression
Lynch syndrome	AD	hMSH2, hMLH1 (hPMS1, hPMS2, hMSH6)	MSH2, MLH1, MSH6, PMS2, PMS1
Familial adenomatous polyposis 1 (FAP)	AD (AR)	APC (MUTYH)	APC (hMYH)
Li-Fraumeni syndrome	AD	TP53	TP53
NF1	AD	NF1	Neurofibromin
TSC-1	AD	TSC1	Hamartin
TSC-2	AD	TSC2	Tuberin
VHL (Von Hippel–Lindau disease)	AD	VHL	pVHL
MEN-1	AD	MEN1	MEN1
MEN-2B	AD	RET	RET
MEN-4	AD	CDKN1B	p27
Polycythemia paraganglioma syndrome	-	EPAS1	HIF2A
Mahvash disease (MVAH)	AR	GCGR	Glucagon receptor

AD: autosomal dominant; AR: autosomal recessive; MEN-1: multiple endocrine neoplasia type I; MEN-2B: multiple endocrine neoplasia type IIB; MEN-4: multiple endocrine neoplasia type IV; NF1: neurofibromatosis type I; TSC-1: tuberous sclerosis complex type I; TSC-2: tuberous sclerosis complex type II.

**Table 3 cancers-14-02028-t003:** The sensitivity, advantages and disadvantages of different imaging methods in GEP-NEN patient diagnosis.

Heading	Sensitivity	Advantages	Disadvantages
US	PanNEN 13–27% 85–90% liver metastases (CEUS—99%)	Cheap, widely accessible Can be enhanced by using contrast	Not recommended for the other parts of the GI tract
EUS	PanNET 54–97% Insulinomas 71–94%	Sensitivity of insulinoma detection higher than that achieved using CT (20–63%) Possibility of fine needle biopsy of the lesion	The quality of the test depends greatly on the skill of the person performing the test Possibility of mistaking high-grade NENs for adenocarcinoma
CT contrast enhanced	PanNET 63–82% Metastases to the liver 82%, lymph nodes 60–70%, bones 58%	Good visualization of vascular infiltration	Low sensitivity for detecting lesions < 1 cm and small lesions in duodenum, stomach and small intestine and bone metastases
MRI contrast enhanced	PanNEN 79%	Less radiation to the patient than that in CT	Low sensitivity for detecting small lesions in the duodenum, stomach and small intestine
DWI	83% liver metastases	Detection of lesions after treatment Easier differentiation of hepatic NEN metastases from hemangiomas Measures the effectiveness of PRRT	
^111^In-pentetreotide SPECT/CT	PanNET 60–80% 90–100% liver metastases	Better sensitivity than standard methods	More radiation Less effective than ^68^Ga-DOTA
^68^Ga-DOTA-SSA PET/CT	PanNETs 79.6% 95–100% liver metastases (DOTATAC)	Less radiation to the patient The most effective method listed	Low half-life time—68 min

CEUS: contrast-enhanced US; DWI: diffusion-weighted imaging; EUS: endoscopic US; GI: gastrointestinal; NEN: neuroendocrine neoplasm; PanNET: pancreatic neuroendocrine tumor; PanNEN: pancreatic neuroendocrine neoplasm; PRRT: peptide receptor radionuclide therapy; SSA: somatostatin analogs.

**Table 4 cancers-14-02028-t004:** Treatment regimen for GEP NETs and GEP NECs.

Locoregional/Resectable	Metastatic/Unresectable									
G1 G2 G3	G1/G2							G3		
Surgery	Asymptomatic/stable	Symptomatic/progressive						NET	NEC	
	Observe SSA	Hepatic dominant/metastases	Widespread					Cisplatin/Eotpside Everolimus Sunitinib ** Platinum/Etopside CAP/TEM PRRT SSTR- rapid SSTR+ slow Studies	Ki-67 50–60%	Ki-67 >60%
									Platinum/Etoposide CAP/TEM FOLFIRI FOLFOX Studies	Plapinum/Etopside FOLFIRI FOLFOX Studies
		Chemoembolization STZ	panNET			Midgut NET				
			SSTR+	SSTR+ Ki-67 >10%	SSTR-	SSTR+	SSTR-			
			Everolimus Studies Sunitinib Lanreotide SSA PRRT	STZ/FU CAP/TEM *	Everolimus Studies Sunitinib STZ/FU CAP/TEM	Everolimus IFN alpha Studies Octreotide Lanreotide SSA PRRT	Everolimus IFN alpha Studies			

* STZ/FU, CAP/TEM- if progression occurs after treatment, continue treatment as recommended for SSTR+, except for the use of SSA. ** Sunitinib only for PanNET. CAP: capecitabine; Ki-67: rate of cell growth; TEM: temozolomide; FOLFIRI: treatment regimen that includes folinic acid, fluorouracil and irinotecan hydrochloride; FOLFOX: treatment regimen that includes folinic acid, fluorouracil and oxaliplatin, FU: fluorouracil; PRRT: peptide receptor radionuclide therapy; SSA: somatostatin analogs; SSTR: somatostatin receptor; STZ: streptozocin.

## Data Availability

The datasets used and/or analyzed during the current study are available from the corresponding author upon reasonable request.

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
