# Peer review of "Advances in the Diagnosis and Therapeutic Management of Gastroenteropancreatic Neuroendocrine Neoplasms (GEP-NENs)"

_cancers, 2022, doi:10.3390/cancers14082028_

Round 1

Reviewer 1 Report

The authors described a manuscript entitled “

Advances in the Diagnosis and Therapeutic Management of Gastroenteropancreatic Neuroendocrine Tumors (GEP-NETs)” as a review. It contains topics ranging from diagnosis and therapeutics, comprehensively.

In the part of “Surgery”, the information might be less enough in some part. For example,

5.1. Stomach

ENETS guideline recommend that tumors with 1cm< are indication for surgical resection (Although NCCN guideline does not clearly mention about the size of tumors). Both guidelines recommend gastrectomy with lymphadenectomy for the tumor with invasion into muscularis propria (mp) or deeper.

5.3. Pancreas

I agree that pancreatic resection with lymphadenectomy is a gold standard procedure for most lesions. Some small tumors with about 1cm in size, especially small insulinoma or non-functioning NET that is found incidentally, is a good candidate for limited resection (enucleation).

Addition of these information would improve the manuscript.

Author Response

Reviewer 1: Dear reviewer, tank you very much for your suggestions. We introduced all of them to the manuscript as follows:

    1. „The authors described a manuscript entitled “Advances in the Diagnosis and Therapeutic Management of Gastroenteropancreatic Neuroendocrine Tumors (GEP-NETs)” as a review. It contains topics ranging from diagnosis and therapeutics, comprehensively.”
    2. Thank you very much for your opinion.
    1. “In the part of “Surgery”, the information might be less enough in some part. For example

5.1. Stomach

ENETS guideline recommend that tumors with 1cm< are indication for surgical resection (Although NCCN guideline does not clearly mention about the size of tumors). Both guidelines recommend gastrectomy with lymphadenectomy for the tumor with invasion into muscularis propria (mp) or deeper.

5.3. Pancreas

I agree that pancreatic resection with lymphadenectomy is a gold standard procedure for most lesions. Some small tumors with about 1cm in size, especially small insulinoma or non-functioning NET that is found incidentally, is a good candidate for limited resection (enucleation). “

  1. Thank You very much for your suggestions.

In the section concerning the stomach we added the information you wrote, and some others like the types of tumor resections.  We added some other references that address these issues and supports these data. To the section of pancreas we added National Comprehensive Cancer Network (NCCN) statement. 

      Thank you very much for this suggestion.

Reviewer 2 Report

The review "Advances in the Diagnosis and Therapeutic Management of 2 Gastroenteropancreatic Neuroendocrine Tumors (GEP-NETs)" is overall nice but not novel. There is really not much new information.  

The English use overall is sub-standard and requires extensive native speaker editing.

The content organization can be not so careful. For example, section 1.2 is named etiology but the content is entirely epidemiology. 

There are also scientific errors. For example, Table 2 is supposed to be about genetic syndromes. The polycytema paraganglioma syndrome is not inherited, however. It is sporadic due to early acquired embryonic mutation. On the other hand, an autosomal recessive hereditary disease (Mahvash disease, OMIM #619290) is not listed.

Author Response

Reviewer 2: Dear reviewer, thank you very much for your opinion, that “The review "Advances in the Diagnosis and Therapeutic Management of  Gastroenteropancreatic Neuroendocrine Tumors (GEP-NETs)" is overall nice …”  It is extremely important for us. Thank you.  

    1. “The review is not novel. There is really not much new information.”
    2. Dear reviewer, in our work we tried to use the most recent sources of information available in data bases, which we searched for the diagnostic and therapeutic methods. We compared the presented novelties with previously used methods in order to compare their effectiveness. Of the 141 papers we cited, as many as 30 of them were published in 2021 and 31 in 2020, what is 43% of all papers used. More than 2/3 of the cited papers were published no later than 2016. The use of the older papers was often intended to present the comparison of the formerly used methods to these new ones, in order to show the progression of the field. We quoted older articles concerning the issues, which were not changed until now, like definitions, symptoms, etc. Compared to the first version of our manuscript, we added some new information and quoted some new articles. We see it improved the quality and novelty of our work. Thank you.
    1. “The English use overall is sub-standard and requires extensive native speaker editing.”
    2. Thank you for this comment. We resubmitted once again our manuscript to Native Speaker, i.e. American Journal Experts for the second revision. Certificate is attached.
    1. “The content organization can be not so careful. For example, section 1.2 is named etiology but the content is entirely epidemiology.” 
    2. Dear reviewer, thank you for this remark. Indeed, this is an error in the article, that was made during editing process. We corrected it by merging the content of Epidemiology paragraph with the content of Etiology We entitled this new one as Epidemiology. Thank you very much for noticing and indicating this mistake.
    1. “There are also scientific errors. For example, Table 2 is supposed to be about genetic syndromes. The polycytema paraganglioma syndrome is not inherited, however. It is sporadic due to early acquired embryonic mutation. On the other hand, an autosomal recessive hereditary disease (Mahvash disease, OMIM #619290) is not listed.”
    2. Dear reviewer, thank you for this remark. An error occured in the article is corrected. We removed the word "inherited" from the title of the table. Our aim, among others, was to indicate the most important mutations and genetic syndromes, including those listed in the WHO Classification, and predisposing to diseases of this group. To form this table, we brought some details from the article “Novel insights into the polycythemia-paraganglioma-somatostatinoma syndrome.” Roland Darr at al., published in Endocrine-Related Cancer in 2016 (https://doi.org/10.1530/ERC-16-0231). In the “Supplementary Table 3” there were some patients with familial mutations in HIF2A gene (with germline mutation) with specific symptoms. Of course typical polycythemia paraganglioma syndrome with enormous prevalence is, as you mentioned, caused by sporadic mutation and, before mentioned cases, are extremely rare. Thank you very much.

Reviewer 3 Report

The manuscript is a comprehensive overview of the GEP-NETS, discussing the newest achievements, and covering all aspects of this field. It is well-organized, the References are up-to-date, the Tables are clear. There are some points that should be corrected before publishing, otherwise I suggest the paper be accepted.

Line 43: the icNET terminology is strange and not accepted for pancreatic NETs, the PanNET is preferable.

Table 2: correct: polycythemia

Line 94-103: for survival data the generally used/accepted formats should be used:  97%, 81 %, 39%, ...etc.

There is confusion about the spelling of the pancreatic NETs. The uniformly used abbreviation is PanNET, not PNET (line 159), not pNET (Table 3). By the way, the spelling must be unified throughout the paper (see lines 540, 542, 699, 704, 891).

Table 3, explanations: neuronendocrineneoplasm (=two words)

Table 4: correctly: capecitabine

Author Response

Reviewer 3: Dear reviewer, tank you very much for your suggestions. We introduced all of them to the manuscript as follows:

    1. „The manuscript is a comprehensive overview of the GEP-NETS, discussing the newest achievements, and covering all aspects of this field. It is well-organized, the References are up-to-date, the Tables are clear. There are some points that should be corrected before publishing, otherwise I suggest the paper be accepted.”
    2. Dear reviewer, thank you very much for your opinion. We have done our best to make the corrections you pointed out.
    1. “Line 43: the icNET terminology is strange and not accepted for pancreatic NETs, the PanNET is preferable.”
    2. We removed the sentence with the use of "icNET". It was previously intended to indicate an example of a disease name, however, indeed, such terminology is not currently used.
    1. “Table 2: correct: polycythemia
    2. There was a misspelling in Table 2, which we corrected.
    1. “Line 94-103: for survival data the generally used/accepted formats should be used:  97%, 81 %, 39%, ...etc.”
    2. Similarly, in the disease prediction section, we have made changes as indicated. Thank you for these comments, they will significantly improve the quality of our article.
    1. “There is confusion about the spelling of the pancreatic NETs. The uniformly used abbreviation is PanNET, not PNET (line 159), not pNET (Table 3). By the way, the spelling must be unified throughout the paper (see lines 540, 542, 699, 704, 891).”
    2. We have standardized the abbreviations corresponding to pancreatic NETs to PanNET throughout the paper, as you suggested. We also changed the nomenclature of PNET to PanNET in Abbreviations. Thank you for bringing this to our attention. Thank you.
    1. “Table 3, explanations: neuronendocrineneoplasm (=two words)”
    2. We corrected the expansion of the abbreviation NEN under the table 3 from “neuroendocrineneoplasm” to “neuroendocrine neoplasm”
    1. “Table 4: correctly: capecitabine”
    2. We corrected the word ‘cabecitabine’ to ‘capecitabine’ in Table 4.

Reviewer 4 Report

Thank you for giving me the opportunity to review the interesting article. The authors summarized the recent advances regarding the diagnosis and treatment of GEP-NENs. Overall, this review is well-written and presented in a timely fashion. I think the different aspects discussed in this review will be informative for readership of the journal because the diagnosis and treatment of GEP-NENs is evolving rapidly. However, there are some minor points that should be addressed. I would like to attach the Reviewer’s comments as below.

  1. Throughout the manuscript the authors used the terms of both ‘GEP-NETs’ and ‘GEP-NENs’. I recommend using the terms appropriately because GEP-NENs includes both NETs and NECs. Since the authors described the treatment of NEC as well as NET, ‘GEP-NENs’ is more appropriate for the title. Please check above carefully throughout the manuscript.

  1. I think there are few descriptions about EUS and EUS-FNA in the manuscript. EUS-FNA plays a crucial role especially in the diagnosis of PanNENs. Sensitivity of imaging modalities depends on tumor size or location. EUS is usually most effective for detecting PanNET, even with small size. I feel that EUS sensitivity of 54% is too low to detect PanNET.

  1. I also recommend adding descriptions about the importance of pathological findings. Among them, Ki-67 index is well known as the most reliable prognostic factor. Please discuss above points.

  1. I think the term ‘Carcinoma syndrome’ is a mistake of ‘Carcinoid syndrome’. Please check it carefully throughout the manuscript.

  1. In Table 4, the term ‘Sunitynib’ is a misspelling of ‘Sunitinib’.

Author Response

Reviewer 4: Dear reviewer, tank you very much for your suggestions. We introduced all of them to the manuscript as follows:

    1. „Thank You for giving me the opportunity to review the interesting article. The authors summarized the recent advances regarding the diagnosis and treatment of GEP-NENs. Overall, this review is well-written and presented in a timely fashion. I think the different aspects discussed in this review will be informative for readership of the journal because the diagnosis and treatment of GEP-NENs is evolving rapidly. However, there are some minor points that should be addressed. I would like to attach the Reviewer’s comments as below.”
    2. Dear reviewer, thank you very much for your warm opinion about our article. Thank you. We have done our best to incorporate your suggestions and corrections to improve the quality of our text.
    1. „Throughout the manuscript the authors used the terms of both ‘GEP-NETs’ and ‘GEP-NENs’. I recommend using the terms appropriately because GEP-NENs includes both NETs and NECs. Since the authors described the treatment of NEC as well as NET, ‘GEP-NENs’ is more appropriate for the title. Please check above carefully throughout the manuscript.”

  1. In the paper we described both NECs and NETs, so we followed your advice and changed the title to the one you suggested. We also changed the naming of GEP-NETs to GEP-NENs in: Simple Summary, Abstract, and Keywords. Additionally, we included additional sentence in Simple Summary: "Special attention was given to GEP-NETs as the most widely reported in the literature." We also carefully reviewed the remainder of the manuscript to check that the nomenclature was used correctly. We have also made changes of the naming of GEP-NETs to GEP-NENs in Conclusions.

Thank you for this advice.

    1. “I think there are few descriptions about EUS and EUS-FNA in the manuscript. EUS-FNA plays a crucial role especially in the diagnosis of PanNENs. Sensitivity of imaging modalities depends on tumor size or location. EUS is usually most effective for detecting PanNET, even with small size. I feel that EUS sensitivity of 54% is too low to detect PanNET.”
    2. Thank you for this advice. We added 18 lines of text in the Imaging Test section elaborating on EUS and EUS-FNA, citing 2 newly added papers to the footnotes from: 2017 "Clinical impact of endoscopic ultrasonography on the management of neuroendocrine tumors: lights and shadows" and 2022 - "Update on Epidemiology, Diagnosis, and Biomarkers in Gastroenteropancreatic Neuroendocrine Neoplasms". Based on Your guidance on the dependence of the diagnostic value on the size and location of the lesion, we evaluated the role of EUS in the diagnosis of GEP-NENs with a special emphasis on Pan-NENs. We also highlighted the role of EUS-FNA for the diagnosis of GEP-NENs with the special look on its role for the diagnosis of PanNENs.
    1. “I also recommend adding descriptions about the importance of pathological findings. Among them, Ki-67 index is well known as the most reliable prognostic factor. Please discuss above points.”
    2. Dear reviewer, we are very grateful for your advice. Indeed, the point you mentioned is animportant and mandatory one. So, we decided to enrich our paper with a section on Ki-67 index.We have done our best to include all the important points in our paper.
    1. “I think the term ‘Carcinoma syndrome’ is a mistake of ‘Carcinoid syndrome’. Please check it carefully throughout the manuscript.”
    2. Of course, the use of the term ‘Carcinoma syndrome’ was a significant lapse. We replaced it with the correct term ‘Carcinoid syndrome’. Thank you for this comment.
    1. “In Table 4, the term ‘Sunitynib’ is a misspelling of ‘Sunitinib’.”
    2. We corrected the name ‘Sunitynib’ for ‘Sunitinib’ as you rightly suggested. Thank you for this remark.

Round 2

Reviewer 2 Report

no more issues